# Factors associated with active syphilis among men and women aged 15 years and older in the Zimbabwe Population-based HIV Impact Assessment (2015–2016)

**Leala Ruangtragool[1], Rachel Silver[1], Anna Machiha[2], Lovemore Gwanzura[3], Avi Hakim[4], Katie Lupoli[4], Godfrey Musuka[5], Hetal Patel[4], Owen Mugurungi[2], Beth A. Tippett Barr[6], John H. Rogers[6]***

1 Public Health Institute / CDC Global HIV Surveillance Fellow, Harare, Zimbabwe, 2 Ministry of Health and Child Care, Harare, Zimbabwe, 3 Biomedical Research and Training Institute, Harare, Zimbabwe, 4 Division of Global HIV/TB, U.S. Centers for Disease Control and Prevention, Atlanta, GA, United States of America, 5 ICAP at Columbia University, Harare, Zimbabwe, 6 Division of Global HIV/TB, U.S. Centers for Disease Control and Prevention, Harare, Zimbabwe

* yet6@cdc.gov

**Data Availability Statement:** The data may be found through ICAP at Columbia University at the

## Abstract

### Introduction

Ulcerative STIs, including syphilis, increase the risk for HIV acquisition and transmission due to the presence of ulcers/chancres that serve as a point-of-entry and exit for HIV. In Zimbabwe, diagnosis of syphilis often occurs in pregnant women who seek ANC services where syphilis testing is offered, and among men and women who seek health care for STIs. Zimbabwe's national syphilis estimates are based on these diagnosed cases, with little information available about the prevalence of untreated syphilis among the general population. This analysis uses data from ZIMPHIA (2015–2016) to describe factors associated with active syphilis among men and women ages 15 years and older.

### Methods

ZIMPHIA collected blood specimens for HIV and syphilis testing from 22,501 consenting individuals (ages 15 years and older). Household HIV testing used the national HIV rapid-testing algorithm with HIV-positive results confirmed at satellite laboratories using Geenius HIV-1/2 rapid test (Bio-rad, Hercules, California, USA). Point-of-care non-Treponemal and Treponemal syphilis testing was performed using Chembio's Dual-Path Platform Syphilis Screen & Confirm Assay. Factors associated with active syphilis were explored using multiple variable, weighted logistic regression and were stratified by gender.

### Results

The likelihood of active syphilis in HIV-positive females was 3.7 times greater in HIV-positive females than HIV-negative females (aOR: 3.7, 95% CI 2.3–5.9). Among males odds of having active syphilis was 5 times higher among those that engaged in transactional sex than

following link: https://phia-data.icap.columbia.edu/datasets?country_id=6.

**Funding:** This publication was supported by the President's Emergency Plan for AIDS Relief (PEPFAR) through CDC to ICAP at Columbia University under the terms of the cooperative agreement #U2GGH001226.

**Competing interests:** The authors have declared that no competing interests exist.

those who did not have sex or transactional sex (aOR: 5.3, 95% CI 1.9–14.7), and 6 times higher if HIV positive versus negative (aOR: 5.9, 95% CI 3.0–12.0). Urban residence, province, education (highest attended), marital status, number of sex partners, consistency of condom use, pregnancy status (females), and circumcision status (males) were not significant in the adjusted model for either females or males.

## Conculsion

HIV status was found to be the only factor associated with active syphilis in both females and males. Given the persistent link between HIV and active syphilis, it is prudent to link individuals' diagnoses and treatments, as recommended by the WHO. Enhanced integration of STI and HIV services in health delivery points such as ANC, reproductive services, or male circumcision clinics, combined with consistent, targeted outreach to high-risk populations and their partners, may assist the MOHCC to eliminate active syphilis in Zimbabwe.

## Introduction

Syphilis is a sexually transmitted infection (STI) that may lay dormant in the body until damaging, disfiguring, and potentially deadly symptoms appear years later. Caused by the bacterium *Treponema pallidum*, syphilis has four stages of disease progression when left untreated. The primary stage is characterized by the appearance of painless sores known as chancres, rashes and/or lesions in the secondary stage, a lack of clinical manifestation in the latent stage, and harmful effects on various organ systems in the tertiary stage [1]. Latent stage syphilis has the potential to last for years, allowing the rare tertiary stage syphilis to appear even 10–30 years after initial infection. Neurosyphilis and ocular syphilis can also occur at any stage of infection. The incubation period of syphilis is 21 days, but symptoms may appear anywhere from 10–90 days after acquisition of the bacteria. The primary mode of syphilis transmission is through sexual contact, though vertical transmission may also occur from mother-to-child. When present in pregnant women, syphilis can be passed on to the unborn child and lead to spontaneous abortion, poor fetal development and fatal outcomes in newborns and infants [2]. Infant death is seen in up to 40% of cases if the infection is left untreated in pregnant women [2]. All stages of infection can be effectively treated with penicillin [1]; therefore, timely diagnosis of syphilis is critical in the prevention of disease progression and transmission.

Persons with ulcerative STIs are at an increased risk for human immunodeficiency virus (HIV) acquisition and transmission due to the presence of chancres that serve as a point of entry and exit for HIV [3–5]. Various studies estimate this risk to be two to five times higher among those infected with other STIs, such as syphilis. When co-infection with HIV occurs, both infections tend to progress more rapidly, with HIV-positive persons presenting more frequently with secondary and tertiary stage syphilis [6], and with higher HIV viral loads and lower CD4 counts [4]. Populations at a higher risk for HIV are similarly at an increased risk for syphilis due to the transmission of disease through sexual contact. In Southern Africa, these high-risk populations include pregnant women, uncircumcised men, and persons who engage in unsafe sex, have multiple sexual partners, and practice transactional sex [7–9].

In Zimbabwe, diagnosis of syphilis often occurs in pregnant women who seek antenatal care (ANC) services where syphilis testing is routinely offered, and among men and women who seek health care for STIs [10]. Historically, Zimbabwe's national syphilis estimates are

based on these diagnosed cases, with little information available about the prevalence of untreated syphilis among the general population of adults. Moreover, the most recent health facility-based national estimates may be overestimates as the lack of resources has limited the use of confirmatory laboratory assays for syphilis [11]. Zimbabwe's minimum package for HIV prevention, care, and treatment includes STI testing and syndromic treatment at all facilities, and specifically, that syphilis rapid tests should be stocked at all healthcare facilities, However, economic difficulties and a significant reduction in healthcare workers over the last several years may be disrupting the country's ability to ensure a constant supply of tests and antibiotics, as well as sufficient personnel to staff such facilities. Therefore, Zimbabwe's MoHCC efforts towards infection control continue to be hampered by the lack of data on syphilis incidence, prevalence, and behaviors and factors related to disease acquisition.

The 2015–2016 Zimbabwe Population-based HIV Impact Assessment (ZIMPHIA) survey was able to address these information gaps surrounding syphilis in the adult Zimbabwean population. ZIMPHIA collected demographic and behavioral data on a representative sample of the adult population aged 15 years and older and tested for HIV and syphilis. These data demonstrated that although Zimbabwe has continued to successfully mitigate syphilis around the country, with the prevalence decreasing from 1.9% in 2000 to 1.5% in 2016 [12], HIV positive individuals continue to be at risk for syphilis. ZIMPHIA data can help to assess the degree that syphilis affects high risk populations in Zimbabwe, as well as the current status of active syphilis in the population as a whole. This analysis uses these data to describe active syphilis among men and women ages 15 years and older and the various factors related to infection, helping to uncover populations with increased likelihood for active syphilis that may have been undetected by the current public health system in Zimbabwe.

## Methods

### Sampling

The Zimbabwe Population-based HIV Impact Assessment Survey is a nationally representative, cross-sectional, household survey conducted across Zimbabwe. Stage one of the two-stage stratified cluster sample design selected 500 enumeration areas from the 2012 Zimbabwe Population Census, using a probability proportional to size method. Stage two sampling randomly selected a sample of households in each enumeration area using an equal probability method with an average of 30 households per cluster (range: 15–60). ZIMPHIA data collection occurred between October 2015 and August 2016.

### Consent and individual interview

Individuals 15 years and above were eligible for survey participation. Inclusion criteria additionally required these individuals to have slept in the selected household the night before the interview, and to have demonstrated cognitive ability to provide consent, permission, or assent. Consent and permission procedures, and individual interviews, were administered electronically using a tablet; assent procedures were administered through paper forms. All procedures and interviews were conducted in English, Shona, or Ndebele based on participant preference. Prior to administering individual interviews, electronic informed consent was obtained for survey participation from individuals aged 16 years and above, and emancipated minors aged 15 years. Written assent was obtained for survey participation from unemancipated individuals aged 15 years if parents or guardians provided electronically documented permission to approach the minor. After completion of the individual interview, written consent or assent for participation was additionally obtained from participants in the biomarker component of the survey.

## Biomarker component

ZIMPHIA collected blood specimens for HIV and syphilis testing from 22,501 consenting individuals (ages 15 years and older). Household HIV testing used the national HIV rapid-testing algorithm of Determine (Abbot, Abbott Park, Illinois, USA), First Response (Premier Medical Corporation Private Limited, Daman, India), and Stat-Pak (Chembio, Medford, New York, USA), with HIV-positive results confirmed at satellite laboratories using Geenius HIV-1/2 rapid test (Bio-rad, Hercules, California, USA). Point-of-care non-Treponemal and Treponemal syphilis testing was performed using Chembio's Dual-Path Platform Syphilis Screen & Confirm Assay to diagnose active syphilis. Participants with a test result reactive to both Treponemal and nontreponemal antibodies were considered to have active syphilis. Results from the household HIV and syphilis tests were returned immediately to the participants and those testing positive were referred for treatment at a health facility of their choice.

## Ethical review

The study protocol and all data collection tools were approved by the Medical Research Council of Zimbabwe (MRCZ), and the Institutional Review Board (IRB) at the U.S. Centers for Disease Control and Prevention (CDC) and Columbia University.

## Statistical analyses

We summarized baseline demographic and behavioral characteristics using unweighted frequencies and weighted percentages. All analyses were conducted using jackknife survey weights which account for sample selection probabilities and are adjusted for nonresponse and noncoverage. Covariates of interest we identified from the literature included HIV-status, sexual behavior (i.e. multiple partnerships, condom use, and transactional sex), marital status, age, and gender. We also examined province of residence, residential setting (urban versus rural), male circumcision status, and pregnancy status.

We used weighted logistic regression to calculate odds ratios and 95% confidence intervals for the associations between the independent variables of interest and the dependent variable of interest—active syphilis. Each independent variable was assessed in separate univariate models to investigate the reduction in -2 log likelihoods using chi-square tests with the appropriate degrees of freedom. Independent variables reducing the model deviance at the $p < 0.05$ level were then added one by one into a multiple variable model until no significant reductions in model deviance occurred. We then constructed separate multiple variable models for each gender, as factors associated with active syphilis in men and women may differ. All analyses were conducted using Stata version 14.2 (StataCorp, College Station, Texas, USA). Details on additional demographic and behavioral questions assessed in 2015–2016 ZIMPHIA, as well as a discussion on the methods and findings, can be found in the ZIMPHIA 2015–2016 Final Report.

## Results

The prevalence of active syphilis was 0.9% among all participants—1.0% among females and 0.7% among males (Table 1). Two-thirds (65.8%) of all participants with syphilis lived in rural areas, with minimal variation between males and females. Among individuals with active syphilis, the majority of both males and females were under the age of 50, with nearly three-quarters (73.8%) of females and over half (69.9%) of males in this age group having active syphilis. Among all females with active syphilis, 45.9% and 41.7% were currently married and formerly married, respectively; a small proportion of females with active syphilis (7.1%) were currently

**Table 1. Active syphilis and selected characteristics by sex, ages 15+; ZIMPHIA 2015–2016.**

| | Total | | | | | Females | | | | | Males | | | | |
|---|---|---|---|---|---|---|---|---|---|---|---|---|---|---|---|
| | No Syphilis | | Syphilis | | p-value | No Syphilis | | Syphilis | | p-value | No Syphilis | | Syphilis | | p-value |
| | n | (%†) | n | (%†) | | n | (%†) | n | (%†) | | n | (%†) | n | (%†) | |
| **Residence** | | | | | 0.9676 | | | | | 0.9018 | | | | | 0.7691 |
| Rural | 16,000 | (65.6) | 159 | (65.8) | | 9,078 | (64.4) | 101 | (63.8) | | 6,922 | (67.0) | 58 | (68.9) | |
| Urban | 6,279 | (34.4) | 63 | (34.2) | | 4,034 | (35.6) | 45 | (36.2) | | 2,245 | (33.0) | 18 | (31.1) | |
| **Province** | | | | | 0.0177 | | | | | 0.0341 | | | | | 0.4866 |
| Bulawayo | 2,047 | (6.0) | 20 | (6.2) | | 1,329 | (6.5) | 14 | (5.9) | | 718 | (5.5) | 6 | (6.8) | |
| Manicaland | 2,470 | (12.8) | 15 | (7.7) | | 1,435 | (13.0) | 10 | (8.2) | | 1,035 | (12.6) | 5 | (6.9) | |
| Mashonaland Central | 2,249 | (8.8) | 10 | (5.0) | | 1,210 | (8.0) | 5 | (4.1) | | 1,039 | (9.6) | 5 | (6.5) | |
| Mashonaland East | 2,164 | (10.8) | 15 | (8.1) | | 1,170 | (10.1) | 8 | (6.5) | | 994 | (11.7) | 7 | (10.7) | |
| Mashonaland West | 2,650 | (12.8) | 28 | (14.0) | | 1,417 | (11.6) | 20 | (16.1) | | 1,233 | (14.2) | 8 | (10.9) | |
| Matabeleland North | 2,148 | (5.1) | 29 | (7.4) | | 1,280 | (5.3) | 20 | (7.8) | | 868 | (4.9) | 9 | (6.9) | |
| Matabeleland South | 1,811 | (5.1) | 37 | (10.4) | | 1,092 | (5.4) | 24 | (10.6) | | 719 | (4.8) | 13 | (10.2) | |
| Midlands | 2,190 | (10.8) | 21 | (11.6) | | 1,354 | (11.5) | 13 | (11.0) | | 836 | (10.1) | 8 | (12.7) | |
| Masvingo | 2,397 | (10.4) | 27 | (12.1) | | 1,452 | (11.0) | 20 | (14.3) | | 945 | (9.7) | 7 | (8.7) | |
| Harare | 2,153 | (17.3) | 20 | (17.3) | | 1,373 | (17.7) | 12 | (15.7) | | 780 | (16.9) | 8 | (19.8) | |
| **Sex** | | | | | 0.0155 | | | | | | | | | | |
| Female | 13,112 | (52.6) | 146 | (61.2) | | | | | | | | | | | |
| Male | 9,167 | (47.4) | 76 | (38.8) | | | | | | | | | | | |
| **Age (years)** | | | | | 0.0006 | | | | | 0.1212 | | | | | 0.0004 |
| 15–24 | 7,061 | (34.2) | 40 | (22.4) | | 3,900 | (32.8) | 31 | (27.4) | | 3,161 | (35.8) | 9 | (14.6) | |
| 25–34 | 5,022 | (25.6) | 51 | (24.9) | | 3,114 | (26.5) | 38 | (27.9) | | 1,908 | (24.7) | 13 | (20.1) | |
| 35–49 | 5,236 | (23.0) | 52 | (25.0) | | 3,110 | (22.2) | 28 | (18.5) | | 2,126 | (23.9) | 24 | (35.2) | |
| 50+ | 4,960 | (17.1) | 79 | (27.7) | | 2,988 | (18.4) | 49 | (26.2) | | 1,972 | (15.6) | 30 | (30.1) | |
| **Education: Highest attended** | | | | | <0.001 | | | | | 0.0015 | | | | | 0.0031 |
| No education | 1,033 | (3.6) | 21 | (7.6) | | 806 | (5.2) | 16 | (9.7) | | 227 | (1.8) | 5 | (4.2) | |
| Primary | 7,304 | (28.2) | 104 | (42.3) | | 4,450 | (30.4) | 68 | (42.2) | | 2,854 | (25.9) | 36 | (42.4) | |
| Secondary or above | 13,916 | (68.2) | 97 | (50.1) | | 7,843 | (64.4) | 62 | (48.1) | | 6,073 | (72.4) | 35 | (53.4) | |
| **Marital status** | | | | | <0.001 | | | | | <0.001 | | | | | <0.001 |
| Never married | 6,080 | (30.1) | 27 | (13.9) | | 2,688 | (22.3) | 16 | (12.4) | | 3,392 | (38.7) | 11 | (16.1) | |
| Married/Living together | 12,712 | (56.5) | 118 | (53.0) | | 7,523 | (57.3) | 67 | (45.9) | | 5,189 | (55.6) | 51 | (64.3) | |
| Formerly married‡ | 3,458 | (13.4) | 76 | (33.1) | | 2,883 | (20.4) | 62 | (41.7) | | 575 | (5.8) | 14 | (19.6) | |
| **No. of sexual partners (last 12 mos)** | | | | | 0.0003 | | | | | <0.001 | | | | | 0.2170 |
| 0 | 4,180 | (21.4) | 47 | (20.3) | | 2,829 | (24.1) | 37 | (25.2) | | 1,351 | (18.3) | 10 | (12.5) | |
| 1 | 12,254 | (68.0) | 122 | (59.4) | | 7,825 | (72.9) | 81 | (59.1) | | 4,429 | (62.3) | 41 | (59.9) | |
| 2+ | 1,638 | (10.5) | 31 | (20.3) | | 315 | (3.0) | 16 | (15.8) | | 1,323 | (19.4) | 15 | (27.7) | |
| **Condom use (last 12 mos)** | | | | | 0.0093 | | | | | <0.001 | | | | | 0.4019 |
| No sex/Always used | 6,970 | (37.1) | 86 | (39.8) | | 4,234 | (36.3) | 63 | (45.1) | | 2,736 | (37.9) | 23 | (31.2) | |
| Inconsistent use | 686 | (4.5) | 14 | (9.6) | | 150 | (1.3) | 7 | (7.6) | | 536 | (8.2) | 7 | (12.8) | |
| Never used | 10,320 | (58.4) | 98 | (50.6) | | 6,555 | (62.3) | 63 | (47.3) | | 3,765 | (53.8) | 35 | (56.0) | |
| **Transactional sex (last 12 mos)‡‡** | | | | | <0.001 | | | | | <0.001 | | | | | <0.001 |
| No sex/No trans. sex | 17,359 | (95.5) | 170 | (81.6) | | 10,631 | (96.9) | 115 | (83.9) | | 6,728 | (94.0) | 55 | (78.0) | |
| Yes, trans. sex | 749 | (4.5) | 30 | (18.4) | | 359 | (3.1) | 19 | (16.1) | | 390 | (6.0) | 11 | (22.0) | |
| **Pregnancy status** | | | | | | | | | | 0.2773 | | | | | |
| Not pregnant | | | | | | 12,295 | (95.0) | 134 | (92.9) | | | | | | |
| Pregnant | | | | | | 599 | (5.0) | 9 | (7.1) | | | | | | |
| **Male circumcision status** | | | | | | | | | | | | | | | 0.4051 |

*(Continued)*

**Table 1.** (Continued)

| | Total | | | | | Females | | | | | Males | | | | |
|---|---|---|---|---|---|---|---|---|---|---|---|---|---|---|---|
| | No Syphilis | | Syphilis | | p-value | No Syphilis | | Syphilis | | p-value | No Syphilis | | Syphilis | | p-value |
| | n | (%†) | n | (%†) | | n | (%†) | n | (%†) | | n | (%†) | n | (%†) | |
| Not circumcised | | | | | | | | | | | 7,495 | (84.5) | 67 | (88.9) | |
| Circumcised | | | | | | | | | | | 1,430 | (15.5) | 7 | (11.1) | |
| **HIV status** | | | | | <0.001 | | | | | <0.001 | | | | | <0.001 |
| Negative | 18,872 | (86.7) | 122 | (54.7) | | 10,892 | (85.1) | 78 | (54.5) | | 7,980 | (88.6) | 44 | (55.0) | |
| Positive | 3,407 | (13.3) | 100 | (45.3) | | 2,220 | (14.9) | 68 | (45.5) | | 1,187 | (11.4) | 32 | (45.0) | |
| **Total** | 22,279 | (100.0) | 222 | (100.0) | | 13,112 | (100.0) | 146 | (100.0) | | 9,167 | (100.0) | 76 | (100.0) | |
| **Active Syphilis Prevalence** | | | | **0.9** | | | | | **1.0** | | | | | **0.7** | |

† Weighted column percentages.

‡ Formerly married includes people who reported being divorced, separated, or widowed at the time surveyed.

‡‡ Transactional sex includes people who reported that in the past 12 months they paid money for sex, sold money for sex, and/or entered into a sexual relationship with a partner who provided or was expected to provide material support to the respondent. Material support was defined as helping the respondent to pay for things or giving the respondent gifts or other items the respondent needed or requested.

pregnant. Most males with active syphilis were currently married (64.3%), while only 19.6% were formerly married. Approximately half of all females (48.1%) and males (53.4%) with active syphilis reported attending secondary school or higher.

As seen in Table 1, multiple sexual partnerships in the year preceding the survey were reported among 15.8% of females and 27.7% of males with active syphilis. A quarter of females with syphilis (25.2%) reported no sexual partners in the last year, compared to only 12.5% of males. Among all participants with active syphilis, about half (50.6%) reported never using a condom in their sexual partnerships in the past year. Among males with active syphilis, only one in nine (11.1%) reported being circumcised. Transactional sex in the past year was prevalent among 16.1% of females with active syphilis and 22.0% of males. Nearly half of females (45.5%) and males (45.0%) with active syphilis were co-infected with HIV.

Crude odds ratios (cORs) and adjusted odds ratios (aORs) from the multiple variable logistic regressions are presented in Table 2. In the unadjusted models, Matabeleland South was the only province that showed significantly greater difference than Bulawayo in odds of active syphilis (cOR: 2.0, 95% CI: 1.2–3.4). Participants 25 years and older had greater odds of having an active syphilis than those ages 15–24 years, though only participants ages 35–49 (cOR: 1.7, 95% CI: 1.0–2.7) and ages 50+ (cOR: 2.5, 95% CI: 1.6–3.9) were significantly associated with active syphilis. Participants that completed secondary education or higher were significantly less likely to have active syphilis, compared to those that had no education (cOR: 0.4, 95% CI: 0.2–0.7). Being married or living together (cOR: 2.0, 95% CI: 1.2–3.5) and being formerly married (cOR: 5.4, 95% CI: 3.1–9.3) were associated with greater odds of having active syphilis compared to those who were never married.

The odds of having active syphilis were twice as high among people who reported multiple sexual partnerships in the past year compared to those who did not have sex (cOR: 2.0, 95% CI: 1.3–3.3), as well as among those who inconsistently used condoms compared to those who always used them or did not engage in sex (cOR: 2.0, 95% CI: 1.1–3.6). Transactional sex in the past year and HIV infection were strongly associated with active syphilis in the unadjusted analyses with crude odds ratios of 4.8 (95% CI: 3.2–7.3) and 5.4 (95% CI: 4.0–7.3), respectively.

In the adjusted models, participants aged 35–49 years (aOR: 0.6, 95% CI 0.3–1.0), having been formerly married (aOR: 2.1, 95% CI: 1.0–4.1), reporting transactional sex in the last 12

**Table 2. Crude and adjusted odds ratios of co-variates of active syphilis by sex, ages 15+; ZIMPHIA 2015/2016.**

| | Total | | Females | | Males | |
|---|---|---|---|---|---|---|
| | cOR (95% CI) | aOR (95% CI) | cOR (95% CI) | aOR (95% CI) | cOR (95% CI) | aOR (95% CI) |
| **Residence** | | | | | | |
| Rural | ref | | ref | | ref | |
| Urban | 1.0 (0.7–1.4) | 0.9 (0.6–1.5) | 1.0 (0.7–1.5) | 1.7 (0.7–2.0) | 0.9 (0.5–1.6) | 0.4 (0.1–1.2) |
| **Province** | | | | | | |
| Bulawayo | ref | | ref | | ref | |
| Manicaland | 0.6 (0.3–1.1) | 0.6 (0.3–1.2) | 0.7 (0.3–1.4) | 0.8 (0.4–1.8) | 0.4 (0.1–1.5) | 0.2 (0.0–1.4) |
| Mashonaland Central | 0.6 (0.3–1.2) | 0.4 (0.2–1.1) | 0.6 (0.2–1.5) | 0.6 (0.2–1.7) | 0.6 (0.2–1.9) | 0.2 (0.0–1.4) |
| Mashonaland East | 0.7 (0.4–1.4) | 0.7 (0.3–1.5) | 0.7 (0.3–1.6) | 1.0 (0.4–2.6) | 0.8 (0.2–2.4) | 0.3 (0.1–1.7) |
| Mashonaland West | 1.1 (0.5–2.1) | 1.0 (0.5–2.2) | 1.5 (0.8–3.0) | 1.8 (0.8–4.1) | 0.6 (0.1–3.1) | 0.4 (0.1–2.4) |
| Matabeleland North | 1.4 (0.8–2.6) | 0.9 (0.4–1.9) | 1.6 (0.9–3.1) | 1.2 (0.6–2.7) | 1.1 (0.3–4.0) | 0.4 (0.1–2.6) |
| Matabeleland South | **2.0* (1.2–3.4)** | 1.2 (0.6–2.5) | **2.2* (1.2–4.0)** | 2.0 (0.9–4.4) | 1.7 (0.6–5.2) | 0.4 (0.1–2.2) |
| Midlands | 1.0 (0.6–1.9) | 1.0 (0.5–2.0) | 1.1 (0.5–2.2) | 1.4 (0.6–3.3) | 1.0 (0.3–3.1) | 0.5 (0.1–2.4) |
| Masvingo | 1.1 (0.7–2.0) | 0.9 (0.5–2.0) | 1.4 (0.8–2.7) | 1.5 (0.7–3.3) | 0.7 (0.2–2.5) | 0.4 (0.1–2.1) |
| Harare | 1.0 (0.5–1.8) | 0.9 (0.5–1.6) | 1.0 (0.4–2.2) | 0.8 (0.3–1.8) | 1.0 (0.3–3.0) | 1.2 (0.3–4.4) |
| **Sex** | | | | | | |
| Females | ref | | | | | |
| Males | **0.7* (0.5–0.9)** | 0.9 (0.6–1.2) | - | - | - | - |
| **Age (years)** | | | | | | |
| 15–24 | Ref | | ref | | ref | |
| 25–34 | 1.5 (0.9–2.4) | 0.6 (0.4–1.0) | 1.3 (0.7–2.2) | 0.7 (0.4–1.1) | 2.0 (0.8–5.2) | 0.7 (0.2–2.9) |
| 35–49 | **1.7* (1.0–2.7)** | **0.6* (0.3–1.0)** | 1.0 (0.6–1.8) | **0.4* (0.2–0.8)** | **3.6* (1.5–8.8)** | 1.1 (0.3–4.6) |
| 50+ | **2.5** (1.6–3.9)** | 1.0 (0.6–1.6) | **1.7* (1.0–2.9)** | 0.9 (0.4–1.7) | **4.7* (1.9–11.5)** | 1.5 (0.4–5.4) |
| **Education: Highest attended** | | | | | | |
| No education | ref | | ref | | ref | |
| Primary | 0.7 (0.4–1.3) | 0.9 (0.4–2.0) | 0.7 (0.4–1.5) | 0.7 (0.3–1.6) | 0.7 (0.2–2.7) | **2.7* (1.4–5.0)** |
| Secondary or above | **0.4* (0.2–0.7)** | 0.6 (0.3–1.3) | **0.4* (0.2–0.8)** | 0.5 (0.2–1.2) | 0.3 (0.1–1.2) | 1.4 (0.8–2.6) |
| **Marital status** | | | | | | |
| Never married | ref | | ref | | ref | |
| Married or living together | **2.0* (1.2–3.5)** | 1.2 (0.7–2.3) | 1.4 (0.7–3.2) | 1.0 (0.4–2.1) | **2.8* (1.3–5.8)** | 1.0 (0.3–2.7) |
| Formerly married (S, D, W) | **5.4** (3.1–9.3)** | **2.1* (1.0–4.1)** | **3.7* (1.6–8.3)** | 1.6 (0.7–3.8) | **8.2** (3.4–19.5)** | 2.4 (0.7–7.8) |
| **No. of sex partners (last 12 mos)** | | | | | | |
| 0 | ref | | ref | | ref | |
| 1 | 0.9 (0.6–1.3) | 1.1 (0.6–2.0) | 0.8 (0.5–1.2) | 1.4 (0.6–2.9) | 1.4 (0.6–3.0) | 0.8 (0.3–2.2) |
| 2+ | **2.0* (1.3–3.3)** | 1.7 (0.7–4.1) | **5.1* (2.7–9.6)** | **3.5* (1.1–11.7)** | 2.1 (0.9–5.1) | 0.6 (0.2–2.4) |
| **Condom use (last 12 mos)** | | | | | | |
| No sex/Always Used | ref | | ref | | ref | |
| Inconsistent Use | **2.0* (1.1–3.6)** | 1.5 (0.6–3.7) | **4.5* (1.9–10.9)** | 1.0 (0.1–6.0) | 1.9 (0.7–5.0) | 2.7 (0.7–10.7) |
| Never Used | 0.8 (0.6–1.1) | 1.4 (0.8–2.3) | **0.6* (0.4–0.9)** | 0.9 (0.4–1.5) | 1.3 (0.7–2.4) | 2.4 (1.0–5.9) |
| **Transactional sex (last 12 mos)** | | | | | | |
| No sex or no trans. sex | ref | | ref | | ref | |
| Yes-trans. sex | **4.8** (3.2–7.3)** | **3.1** (1.7–5.5)** | **5.9** (3.5–10.0)** | 2.1 (0.9–4.6) | 1.3 (0.6–2.7) | **5.3* (1.9–14.7)** |
| **Pregnancy status** | | | | | | |
| Not pregnant | | | ref | | | |
| Currently pregnant | | | 1.4 (0.7–2.8) | 1.8 (0.9–3.6) | | |
| **Male circumcision status** | | | | | | |
| Not circumcised | | | | | ref | |

*(Continued)*

**Table 2.** (Continued)

| | Total | | Females | | Males | |
|---|---|---|---|---|---|---|
| | cOR (95% CI) | aOR (95% CI) | cOR (95% CI) | aOR (95% CI) | cOR (95% CI) | aOR (95% CI) |
| Circumcised | | | | | 0.7 (0.3–1.8) | 0.8 (0.3–2.4) |
| **HIV status** | | | | | | |
| Negative | ref | | ref | | ref | |
| Positive | 5.4** (4.0–7.3) | 4.7** (3.2–6.9) | 4.8** (3.3–6.8) | 3.7** (2.3–5.9) | 6.4** (3.8–10.6) | 5.9** (3.0–12.0) |

*p<0.05

months (aOR: 3.1, 95% CI 1.7–5.5), and those testing positive for HIV (aOR: 4.7, 95% CI 3.2–6.9) were associated with greater likelihood of active syphilis. Urban or rural residence, province, sex, education, gender, number of sex partners, and condom use were not associated with active syphilis in the general adult population. Though significant relationships with active syphilis existed in the unadjusted analyses for individuals ages 50 years and above, reporting secondary education or above, residing in Matabeleland South, being married or living together, and reporting inconsistent condom use, these co-variates no longer affected the odds of active syphilis in the adjusted model.

## Females

In the univariate logistic regressions, residence and pregnancy status were not significantly associated with active syphilis (Table 2). Similar to the total adult population crude analysis, Matabeleland South was the only province with a significant association with syphilis (cOR: 2.2, 95% CI: 1.2–4.0). Females over 50 years of age had nearly twice the odds of having active syphilis as females between 15–24 years (cOR: 1.7, 95% CI: 1.0–2.9). Additionally, females that reported having multiple sexual partners and engaging in transactional sex were found to have significantly greater odds of active syphilis, with crude odds ratios of 5.1 (95% CI: 2.7–9.6) and 5.9 (95% CI: 3.5–10.0), respectively. Compared to females who always used condoms or did not have sex in the last 12 months, inconsistent condom use was associated with 4.5 times greater odds of active syphilis (cOR, 95% CI: 1.9–10.9). Additionally, the odds of having active syphilis were nearly 5 times as high among HIV-positive females than those negative for HIV (cOR: 4.8, 95% CI: 3.3–6.8).

After adjusting for the other factors that demonstrated significant relationships with active syphilis, the odds of syphilis among females between the ages of 35–49 years were found to be 60% less than the odds among those 15–24 years of age (aOR: 0.4, 95% CI 0.2–0.8). Additionally, females that reported multiple sexual partners in the last 12 months had a 3.5 (95% CI: 1.1–11.7) times greater chance of having active syphilis compared to females who did not have any sexual partners. The likelihood of active syphilis in HIV-positive females decreased in the multivariate logistic regression model but remained 3.7 times greater in HIV-positive females than HIV-negative females (aOR: 3.7, 95% CI 2.3–5.9).

Urban residence, province, education (highest attended), marital status, number of sex partners, consistency of condom use, pregnancy status and transactional sex were not significant in the adjusted model for females.

## Males

Crude analysis for males showed that residence, province, education, number of sex partners, condom use, transactional sex, and male circumcision status had no significant association

with active syphilis in the univariate models (Table 2). Compared to males ages 15–24 years, older age was associated with active syphilis for those 35–49 (cOR: 3.6, 95% CI: 1.5–8.8) and 50 + (cOR: 4.7, 95% CI: 1.9–11.5). Marital status was also associated with greater odds of active syphilis with a crude odds ratio of 2.8 (95% CI: 1.3–5.8) for males currently married or living with their partner, and 8.2 (95% CI 3.4–19.5) for males formerly married. Odds of active syphilis was over 6 times higher for HIV-positive males than their HIV-negative counterparts (cOR: 6.4, 95% CI 3.8–10.6).

Males that reported engaging in transactional sex in the past 12 months or that were HIV-positive were found to have strong associations with active syphilis after adjusting for other covariates. The odds of males having active syphilis was 5 times higher among those that engaged in transactional sex than those who did not have sex or who did not have transactional sex (aOR: 5.3, 95% CI 1.9–14.7), and almost 6 times higher if they were HIV positive versus negative (aOR: 5.9, 95% CI 3.0–12.0). In the multiple variable model, there was no difference in odds of active syphilis among males of different age groups, marital status, provinces, residence types, sex partner or condom use behaviors, or male circumcision status.

## Discussion

We sought to identify factors, behaviors, and high-risk populations associated with active syphilis among Zimbabweans aged 15 years and older. These findings will allow for the development of data-driven, targeted, and cost-effective strategies to assist the Ministry of Health and Child Care in achieving their goal of eliminating syphilis in Zimbabwe by 2030. Compared with other African countries, Zimbabwe demonstrates lower prevalence of active syphilis. The Zambia PHIA found that three percent of persons aged 15–59 years were infected with active syphilis [13] compared to 0.9% in this study. Our results are similar to the active syphilis prevalence in Rwanda (0.9%) [14], which is a country that is close to achieving the UNAIDS 90/90/90 FastTrack goals indicating progress towards HIV epidemic control may correlate with reductions in active syphilis. Though analysis of these ZIMPHIA data further support the decreasing trend in active syphilis prevalence in Zimbabwe and across sub-Saharan Africa [15, 16], efforts towards surveillance, treatment, and STI prevention should remain critical activities in the public health strategy for reducing active syphiliss.

### HIV and syphilis comorbidity

HIV status was found to be the only factor associated with active syphilis in both females and males at p<0.001. Given the persistent link between HIV and active syphilis, it is prudent to link individuals' diagnoses and treatments, as recommended by the World Health Organization [8]. This observation aligns with the findings from Kilmarx and colleagues which showed a HIV prevalence of 41% in patients attending STI clinics in Zimbabwe [17]. Integration of STI and HIV services at all HIV and health service delivery points in Zimbabwe is in progress, but not yet complete at all health facilities. Universal integration of services, specifically the establishment of testing for STIs in concordance with positive HIV diagnoses (and vice versa), could improve detection of both STIs and HIV, leading to earlier treatment, better outcomes, and a reduction in the cost of medical treatment for advanced infections. This would not only provide a key opportunity for early HIV detection, but also for linkage to care for those living with HIV that are not currently on treatment. Integration of prevention and awareness activities, such as education and condom distribution, should additionally be implemented at all STI and HIV testing facilities.

The implementation of innovative technologies like dual HIV-syphilis tests and combined syphilis screen and confirm tests in health facilities, such as the assay used in ZIMPHIA, will

also expand diagnoses and in turn, more timely treatment of syphilis. Though pregnancy status in females, male circumcision, and condom usage were not significantly associated with active syphilis in this analysis, these results should not imply a reduction in prevention activities or programs that promote syphilis screening or informational awareness for these high-risk groups. It is demonstrated throughout the literature that these groups remain highly affected, and these activities are likewise effective in preventing the spread of HIV [4, 6, 7].

### Differences by age

The lack of difference in syphilis between the oldest and youngest age groups is important to note as the focus of STI programming in Zimbabwe is on the vulnerability of young people and others of reproductive age [10]. HIV co-infection could be a reason for the similar probabilities of syphilis in the oldest and youngest age groups. ZIMPHIA results released in August 2019 established that the burden of HIV remained high in 50+ year olds for males and females [18]. Expansion of STI prevention programs and syphilis screening and treatment to the older population whenever they access healthcare for any reason may help to reduce active syphilis in this population.

### Differences between females and males

Active syphilis was found to be higher in females than males, with each sex demonstrating varying associations in the multivariate models, only sharing a significant association between HIV status and active syphilis. A similar trend of active syphilis prevalence was found in the Zambia PHIA [13]. The riskiest behaviors among females and males that demonstrated significant associations with syphilis were engagement in sex with two or more partners for females and transactional sex for males. However, neither of these activities were found to be a significant risk factor for the opposite sex.

Females engaging in sexual activities with two or more partners were at a 3.5 times higher risk of active syphilis than those who did not report having any sex partners. While having two or more sex partners is considered a common risk factor in the acquisition of STI's irrespective of sex, these results further support Celentano et al. findings that in Zimbabwe, the number of sexual partnerships is a factor in having a STI, specifically in females [19].

Males reporting transactional sex were at a 5.3 times higher risk of active syphilis than those who did not have transactional sex or any sex. Chadambuka et al. have noted that in Zimbabwe, men pursue partnerships with commercial sex workers and casual relationships more often than women [20], establishing female sex workers and their male partners as populations vulnerable to STIs, such as syphilis and HIV.

These data additionally revealed that 47.3% of females and 53.8% of males with active syphilis, and 62.3% of females and 53.8% of males without syphilis, reported never using a condom. These results demonstrate a need for targeted testing, treatment, and awareness campaigns aimed at condom usage for both males and females, and specifically at sex workers and the areas they frequent.

### Limitations

The primary objective of 2015–2016 ZIMPHIA was to understand the burden of HIV at subnational levels in Zimbabwe. As such, the study was not powered to detect prevalence of active syphilis at less than a national level, or of the other characteristics of interest. The study results were additionally hindered by missing data from nonresponse, potentially reducing the precision and limiting the ability to make inferences about a given factor or outcome. Despite this limitation of survey design, we believe that these findings still contribute to a baseline, sex-

disaggregated understanding of syphilis in Zimbabwe which is actionable by our partners and the MoHCC.

Secondly, the study results may have been affected by reporting bias for some of the risk-behavior questions, such as condom use, transactional sex, and number of sex partners, as these questions can be considered sensitive in nature and allow for social acceptability bias to occur. Lastly, the findings and recommendations in this study would have less utility if commodities to diagnose and treat syphilis are not consistently available in Zimbabwe.

## Conclusion

The results from this study provide a deeper understanding of the population level burden of syphilis in Zimbabwe. Using multivariable logistic regression, factors associated with increased odds of active syphilis in women were found to include having two or more sex partners in the past 12 months and being HIV positive. In men, these factors included engaging in transactional sex and being HIV positive. Programmatic decisions based on aggregated data from health care facilities or ANC services alone miss key populations that are not seeking treatment or care that therefore remain undetected in active syphilis estimates. This paper demonstrates that although efforts towards reducing the prevalence of syphilis in Zimbabwe have been successful, people living with HIV remain at a higher risk of syphilis, thus increasing their vulnerability to developing an advanced HIV infection. Enhanced integration of STI and HIV services, as well as at other health delivery points such as antenatal care, reproductive services, or male circumcision clinics, combined with consistent and targeted outreach to high-risk populations and their partners, may assist the Ministry of Health and Child Care to more expeditiously eliminate active syphilis in Zimbabwe.

## Acknowledgments

We would like to acknowledge the contributions of the ZIMPHIA Study Team especially the Ministry of Health and Child Care in Zimbabwe and the National AIDS Council of Zimbabwe.

**Disclaimer:** The findings and conclusions in this publication are those of the authors and do not necessarily represent the official position of the funding agencies or any organization represented.

## Author Contributions

**Conceptualization:** John H. Rogers.

**Formal analysis:** Leala Ruangtragool, John H. Rogers.

**Project administration:** Owen Mugurungi.

**Writing – original draft:** Leala Ruangtragool, John H. Rogers.

**Writing – review & editing:** Leala Ruangtragool, Rachel Silver, Anna Machiha, Lovemore Gwanzura, Avi Hakim, Katie Lupoli, Godfrey Musuka, Hetal Patel, Owen Mugurungi, Beth A. Tippett Barr, John H. Rogers.

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
