## [Decision Letter · Decision Letter 0]

29 Apr 2021

PONE-D-20-35608

Factors associated with active syphilis infection among men and women aged 15 years and older in the Zimbabwe Population-based HIV Impact Assessment (2015-2016)

PLOS ONE

Dear Dr. Rogers,

Thank you for submitting your manuscript to PLOS ONE. After careful consideration, we feel that it has merit but does not fully meet PLOS ONE’s publication criteria as it currently stands. Therefore, we invite you to submit a revised version of the manuscript that addresses the points raised during the review process.

Please take note of the reviewers' comments and revise your manuscript in line with their suggestions.

We look forward to receiving your revised manuscript.

Kind regards,

Remco PH Peters, MD, PhD, DLSHTM

Academic Editor

PLOS ONE

Journal Requirements:

3)  We note that the grant information you provided in the ‘Funding Information’ and ‘Financial Disclosure’ sections do not match.

Reviewers' comments:

Reviewer's Responses to Questions

**Comments to the Author**

1. Is the manuscript technically sound, and do the data support the conclusions?

Reviewer #1: Yes

Reviewer #2: Yes

2. Has the statistical analysis been performed appropriately and rigorously? 

Reviewer #1: Yes

Reviewer #2: Yes

3. Have the authors made all data underlying the findings in their manuscript fully available?

Reviewer #1: Yes

Reviewer #2: Yes

4. Is the manuscript presented in an intelligible fashion and written in standard English?

Reviewer #1: Yes

Reviewer #2: Yes

5. Review Comments to the Author

Reviewer #1: This manuscript reports on prevalence of HIV and active syphilis in a large household survey in Zimbabwe using a nationally representative two-stage stratified cluster sample design enrolling 22,501 consenting individuals aged 15 years and older. The methods appear to be sound; the results are based on a large number of observations and well-presented, and the conclusions are supported by the data. Surveys providing unbiased, population-based estimates on the prevalence of and factors associated with two of the most important STIs, i.e., HIV and syphilis, are obviously very important for the planning of STI prevention and care policies locally, while the study methods (as well as study findings) are also relevant for other countries in the region and beyond.

I have the following comments:

1. While generally well-written, the paper is overly long. A word count is not presented, but it appears to be well over the 3,000 word limit that is generally accepted for communications like this. Specifically, the Introduction does not need an extensive description of syphilis morbidity and the Results section is basically a verbatim recap of what’s already in the Tables and can be reduced to some of the highlights that are subsequently considered in the Discussion section. In that context, I have often wondered what the use is of repeating the 95% confidence intervals in the text, which, in my mind just impedes the narrative.

2. The Methods section, though, is short on details. Specifically, is suggests that HIV and syphilis testing were both done in the household settings and that results were given in that setting as well. However, there is no detail on what happened to study participants with positive results. Were they referred for clinical services? Was there follow up that treatment was assured? After all, some of these patients had active syphilis (and some may have had newly diagnosed HIV) with important consequences both clinically and epidemiologically for the prevention of ongoing transmission.

3. The study employs Chembio’s Dual-Path Platform Syphilis Screen & Confirm test and defines “active syphilis” as positive results for both the treponemal and non-treponemal components of the test. However, there is no discussion on the limitations of this test, specifically the low sensitivity for both components in persons with low-titer non-treponemal test results by RPR (see for example: http://dx.doi.org/10.1136/sextrans-2018-053722). It can be argued that, from a clinical perspective it is particularly important to identify those syphilis patients with high titer non-treponemal test results as they have active disease (see for example: https://www.ncbi.nlm.nih.gov/pmc/articles/pmid/31008842/). However, from an epidemiological point of view (i.e., measuring prevalence and incidence), the sensitivity issue is more problematic, especially when comparing data over sequential surveys that may have used different technologies. At the least, this should be discussed as a limitation.

4. As a minor point, I take issue with the term “syphilis infection”, as an infection is not caused by its outcome. Similarly, we say: “gonococcal infection” and not “gonorrhea infection”. Therefore, keep it simply to “syphilis”

Reviewer #2: This is an important manuscript with provide evidence for burden of active syphilis among general population in Zimbabwe. Manuscript not only provide important evidence to estimate burden for national level but also has important programmatic implications to address STI issues in the study country. To improve its readability for international readers, please address the following comments.

Major comments:

- Add findings related to overall burden of active syphilis by gender in results section of abstract. Authors highlighted that PHIA findings would also be beneficial for addressing the bias in syphilis estimates using ANC data and without highlighting burden of syphilis by gender, the findings related to association only is less important.

- Add separate section in methods to specify outcome of interest (outcome variables) so that it would be easy to follow for readers.

- Is it possible to present findings related to treatment status among pregnant women diagnosed with active syphilis? Also highlight prevalence of untreated syphilis by gender (is there any questions that asked ‘whether they seek any STI services in past etc)?

- In ethical consideration, please write what study team done to study participants who diagnosed with HIV positive and active syphilis? (referred for available treatment free of cost or with cost etc.).

- Write briefly about type of STI services in the country and its coverage in introduction section. Is it free or out of pocket payment etc? Try to link study findings with the availability of STI services in the country in discussion section of the manuscript.

- Two-thirds (65.8%) of all participants with syphilis lived in rural areas in country. This is an important finding. Please discuss it in discussion section.

Minor comments:

- Add full form of ZIMPHIA and MOHCC in abstract.

- Be consistent with full form or abbreviation. For ex., Ministry of Health and Child Care. I would recommend using full form for unfamiliar abbreviation.

6. PLOS authors have the option to publish the peer review history of their article (what does this mean?). If published, this will include your full peer review and any attached files.

Reviewer #1: No

Reviewer #2: **Yes: **Dr Keshab Deuba

---

## [Author Response · Author response to Decision Letter 0]

28 Oct 2021

Reviewer #1:

1. While generally well-written, the paper is overly long. A word count is not presented, but it appears to be well over the 3,000 word limit that is generally accepted for communications like this. Specifically, the Introduction does not need an extensive description of syphilis morbidity and the Results section is basically a verbatim recap of what’s already in the Tables and can be reduced to some of the highlights that are subsequently considered in the Discussion section. In that context, I have often wondered what the use is of repeating the 95% confidence intervals in the text, which, in my mind just impedes the narrative.

Respectfully, PLOS ONE style requirements allow for manuscripts of any length. The body of the manuscript is approximately 3,300 words, which when considering the inclusion of measures of association in the body of the Results, is quite close to the referenced 3,000 word limit. The inclusion of 95% confidence intervals is a stylistic preference. We have elected to keep them in the body of the manuscript in order to provide the necessary statistical context of reported measures of association. Doing so keeps the reader from having to flip back and forth between the text and the tables and, from my perspective, aids readability. If the reviewer feels strongly that the introductory paragraph describing the phases of syphilis should be removed, then we are open to that suggestion, but its inclusion is intentional to set the stage for the rest of the manuscript by understanding the morbidity of syphilis.

2. The Methods section, though, is short on details. Specifically, is suggests that HIV and syphilis testing were both done in the household settings and that results were given in that setting as well. However, there is no detail on what happened to study participants with positive results. Were they referred for clinical services? Was there follow up that treatment was assured? After all, some of these patients had active syphilis (and some may have had newly diagnosed HIV) with important consequences both clinically and epidemiologically for the prevention of ongoing transmission.

Addressed in methods.

3. The study employs Chembio’s Dual-Path Platform Syphilis Screen & Confirm test and defines “active syphilis” as positive results for both the treponemal and non-treponemal components of the test. However, there is no discussion on the limitations of this test, specifically the low sensitivity for both components in persons with low-titer non-treponemal test results by RPR (see for example: http://dx.doi.org/10.1136/sextrans-2018-053722). It can be argued that, from a clinical perspective it is particularly important to identify those syphilis patients with high titer non-treponemal test results as they have active disease (see for example: https://www.ncbi.nlm.nih.gov/pmc/articles/pmid/31008842/). However, from an epidemiological point of view (i.e., measuring prevalence and incidence), the sensitivity issue is more problematic, especially when comparing data over sequential surveys that may have used different technologies. At the least, this should be discussed as a limitation.

Thank you for this comment. The manufacturer’s website claims 96.5% sensitivity for the treponemal component and 99.7% sensitivity for the non-treponemal component (for RPR titer >= 1:8). Two published evaluations also demonstrate good performance of the trep line compared to TPPA and the non-trep line compared to RPR/TRUST at titers >= 1:4 but report lower sensitivities in the non-trep line at titers of 1:2 or less (https://www.ncbi.nlm.nih.gov/pmc/articles/PMC3657488/ and https://www.ncbi.nlm.nih.gov/pmc/articles/PMC3008492/pdf/0624-10.pdf). We could state this reported lower sensitivity of the non-trep line of the DPP Screen and Confirm test in specimens with a titer of 1:2 or less as a limitation of the survey methods, but question if it is necessary given the cited length of the manuscript and that a referenced work in PLOS ONE using the same methods (see https://www.ncbi.nlm.nih.gov/pmc/articles/PMC7380641/) did not cite this limitation.

4. As a minor point, I take issue with the term “syphilis infection”, as an infection is not caused by its outcome. Similarly, we say: “gonococcal infection” and not “gonorrhea infection”. Therefore, keep it simply to “syphilis”.

Addressed per recommendation throughout the manuscript.

Reviewer #2:

1. Add findings related to overall burden of active syphilis by gender in results section of abstract. Authors highlighted that PHIA findings would also be beneficial for addressing the bias in syphilis estimates using ANC data and without highlighting burden of syphilis by gender, the findings related to association only is less important.

Thank you for your comment. The prevalence of active syphilis by gender is presented in the first paragraph of the Results.

2. Add separate section in methods to specify outcome of interest (outcome variables) so that it would be easy to follow for readers.

The second paragraph of the statistical analyses section describes both the independent variables of interest and the dependent, or “outcome”, variable, which is active syphilis. 

3. Is it possible to present findings related to treatment status among pregnant women diagnosed with active syphilis? Also highlight prevalence of untreated syphilis by gender (is there any questions that asked ‘whether they seek any STI services in past etc)?

Thank you for this comment. Active syphilis does not differ by pregnancy status. Previous care seeking and testing behavior for HIV is included in the questionnaire as this is primarily an HIV/AIDS survey, but specific questions on care seeking behavior for other STI are not included.

4. In ethical consideration, please write what study team done to study participants who diagnosed with HIV positive and active syphilis? (referred for available treatment free of cost or with cost etc.).

Addressed per comment from Reviewer #1.

5. Write briefly about type of STI services in the country and its coverage in introduction section. Is it free or out of pocket payment etc? Try to link study findings with the availability of STI services in the country in discussion section of the manuscript.

Addressed in Introduction.

6. Two-thirds (65.8%) of all participants with syphilis lived in rural areas in country. This is an important finding. Please discuss it in discussion section.

Thank you for this comment. The significance of this finding is presented in Table 2. This is something the larger PHIA study team has noted and explored for several variables not just syphilis. While true that almost two-thirds of syphilis infections were in rural areas, this result is not significant in our analysis because of the population distribution of Zimbabwe I.e. urban populations are almost entirely limited to Harare and Bulawayo which account for a relatively small percentage of the total population.

---

## [Editor Report · Decision Letter 1]

24 Nov 2021

Factors associated with active syphilis among men and women aged 15 years and older in the Zimbabwe Population-based HIV Impact Assessment (2015-2016)

PONE-D-20-35608R1

Dear Dr. Rogers,

We’re pleased to inform you that your manuscript has been judged scientifically suitable for publication and will be formally accepted for publication once it meets all outstanding technical requirements.

Kind regards,

Remco PH Peters, MD, PhD, DLSHTM

Academic Editor

PLOS ONE
---

## [Editor Report · Acceptance letter]

31 Dec 2021

PONE-D-20-35608R1 

Factors associated with active syphilis among men and women aged 15 years and older in the Zimbabwe Population-based HIV Impact Assessment (2015-2016) 

Dear Dr. Rogers:

I'm pleased to inform you that your manuscript has been deemed suitable for publication in PLOS ONE. Congratulations! Your manuscript is now with our production department. 

Kind regards, 

on behalf of

Prof Remco PH Peters 

Academic Editor

PLOS ONE